# Competence remodels the pneumococcal cell wall exposing key surface virulence factors that mediate increased host adherence

Vikrant Minhas[1,☯], Arnau Domenech[1,☯], Dimitra Synefiaridou[1,☯], Daniel Straume[2], Max Brendel[3], Gonzalo Cebrero[4], Xue Liu[1,5], Charlotte Costa[6], Mara Baldry[6], Jean-Claude Sirard[6], Camilo Perez[4], Nicolas Gisch[7], Sven Hammerschmidt[3], Leiv Sigve Håvarstein[2]*, Jan-Willem Veening[1]*

1 Department of Fundamental Microbiology, Faculty of Biology and Medicine, University of Lausanne, Biophore Building, Lausanne, Switzerland, 2 Department of Chemistry, Biotechnology and Food Science, Norwegian University of Life Sciences, Ås, Norway, 3 Department of Molecular Genetics and Infection Biology, Interfaculty Institute for Genetics and Functional Genomics, Center for Functional Genomics of Microbes, Universität Greifswald, Greifswald, Germany, 4 Biozentrum, University of Basel, Basel, Switzerland, 5 Guangdong Key Laboratory for Genome Stability and Human Disease Prevention, Department of Pharmacology, International Cancer Center, Shenzhen University Health Science Center, Shenzhen, China, 6 Univ. Lille, CNRS, Inserm, CHU Lille, Institut Pasteur Lille, U1019 - UMR 9017 - CIIL - Center for Infection and Immunity of Lille, Lille, France, 7 Division of Bioanalytical Chemistry, Priority Area Infections, Research Center Borstel, Leibniz Lung Center, Borstel, Germany

☯ These authors contributed equally to this work.
* sigve.havarstein@nmbu.no (LSH); jan-willem.veening@unil.ch (J-WV)

**Data Availability Statement:** All relevant data are within the paper and its Supporting information files. The fastq files generated from sequencing are

## Abstract

Competence development in the human pathogen *Streptococcus pneumoniae* controls several features such as genetic transformation, biofilm formation, and virulence. Competent bacteria produce so-called "fratricins" such as CbpD that kill noncompetent siblings by cleaving peptidoglycan (PGN). CbpD is a choline-binding protein (CBP) that binds to phosphorylcholine residues found on wall and lipoteichoic acids (WTA and LTA) that together with PGN are major constituents of the pneumococcal cell wall. Competent pneumococci are protected against fratricide by producing the immunity protein ComM. How competence and fratricide contribute to virulence is unknown. Here, using a genome-wide CRISPRi-seq screen, we show that genes involved in teichoic acid (TA) biosynthesis are essential during competence. We demonstrate that LytR is the major enzyme mediating the final step in WTA formation, and that, together with ComM, is essential for immunity against CbpD. Importantly, we show that key virulence factors PspA and PspC become more surface-exposed at midcell during competence, in a CbpD-dependent manner. Together, our work supports a model in which activation of competence is crucial for host adherence by increased surface exposure of its various CBPs.

available on NCBI with accession number
PRJNA841864.

**Funding:** Work in the Veening lab is supported by
the Swiss National Science Foundation (SNSF)
(project grants 310030_200792 and
310030_192517 to JWV), SNSF JPIAMR grant
(40AR40_185533 to JWV), SNSF NCCR
'AntiResist' (51NF40_180541 to JWV) and ERC
consolidator grant 771534-PneumoCaTChER to
JWV. This work was further supported by grants of
the Deutsche Forschungsgemeinschaft (GI 979/1-2
to NG, HA 3125/5-2 to SH). Work in the Perez lab
is supported by the SNSF (PP00P3_198903 to CP)
and the Helmut Horten Stiftung (HHS to CP)
grants. The funders had no role in study design,
data collection and analysis, decision to publish, or
preparation of the manuscript.

**Competing interests:** The authors have declared
that no competing interests exist.

**Abbreviations:** CBP, choline-binding protein;
CHAP, cysteine/histidine-dependent
amidohydrolase-peptidase; CSP, competence-
stimulating peptide; DMEM, Dulbecco's Modified
Eagle Medium; FCS, fetal calf serum; LTA,
lipoteichoic acid; PFU, plaque-forming unit; PGN,
peptidoglycan; SEM, standard error of the mean;
sgRNA, single-guide RNA; TA, teichoic acid; TSS,
transcription start site; WTA, wall teichoic acid.

## Introduction

*Streptococcus pneumoniae* (the pneumococcus) is a member of the commensal microbiota of
the human nasopharynx. However, it is a major public health problem because in a small pro-
portion of carriers, which nevertheless translates into globally significant numbers, it can cause
severe life-threatening infections such as sepsis, pneumonia, and meningitis [1]. The pneumo-
coccus is a naturally transformable bacterium that can take up and assimilate exogenous DNA
[2–5]. This phenomenon is an important mechanism of genome plasticity and is largely
responsible for the acquisition and spread of antibiotic resistance as well as virulence factors
such as the capsule [6,7]. The transformation process in the pneumococcus requires the induc-
tion of a physiological state named competence, which involves about 10% of the pneumococ-
cal genome [8,9].

Competence is induced by a classical two-component quorum-sensing system in which the
*comC*-encoded competence-stimulating peptide (CSP) is cleaved and exported by the mem-
brane transporter ComAB to the extracellular space [10,11]. Upon a certain threshold of CSP
accumulation, it stimulates the autophosphorylation of the membrane-bound histidine-kinase
ComD, which subsequently activates its cognate response regulator ComE [12–14]. Phosphor-
ylated ComE induces a positive feedback loop to activate the early *com* genes including its own
operon. One of the genes regulated by ComE, *comX*, encodes a sigma factor that activates the
late *com* genes required for DNA repair, DNA uptake, and transformation [13,15]. Although
competence is mostly associated with DNA transformation, only 22 of the genes induced dur-
ing competence are related to transformation [16,17]. Other competence genes are involved in
biofilm formation [18], bacteriocin production [19,20], and siblings' fratricide [21].

As pneumococci do not discriminate between homologous and foreign DNA, it is believed
that fratricide serves to increase and facilitate the exchange of DNA between pneumococci.
Thus, competent pneumococci lyse and subsequently release nutrients and DNA from a sub-
fraction of the population that does not become competent [21]. Three choline-binding pro-
teins (CBPs) constitute the lysis mechanism: CbpD, LytA, and LytC. CbpD is the main driver
of fratricide as the process cannot commence in its absence [22,23]. CBPs are non-covalently
bound to the phosphorylcholine (*P*Cho) of the teichoic acid (TA) [24,25]. CbpD, LytA, and
LytC bind to the *P*Cho residues in the wall teichoic acids (WTAs) that are covalently attached
to the peptidoglycan (PGN) or to the lipid-anchored lipoteichoic acids (LTAs) [23,26,27]. As
most non-pneumococci do not contain choline in their cell walls, lysis by fratricins enhances
the probability for accessing homologous DNA [28]. The pneumococcus is special in its TA
biosynthesis pathway, as the same substrate is used for LTA and WTA (Fig 1A) [29]. It was
recently shown that the lipoteichoic acid ligase TacL is responsible for producing LTA by
transferring the polymeric TA chain from an undecaprenyl-diphosphate (C55-PP) linked pre-
cursor onto the glycolipid anchor [30,31]. The 3 putative enzymes of the LCP-protein family
Cps2A, Psr, and LytR are predicted to anchor the polymeric TA chain from the same precursor
to the PGN to form WTA, but it remains unclear which is the dominant route and how the lev-
els of WTA and LTA are controlled (Fig 1B) [29,32,33]. In addition, it is assumed that most
pneumococcal strains—as with other gram-positive bacteria—anchor their exopolysaccharide
capsule on the same residue within the PGN as WTA, that would make a cross-talk between
these 2 processes likely [32,34]. By contrast, a different attachment position for the capsule in
*S. pneumoniae* has been suggested recently [35] but final structural proof on the molecular
level, e.g., by 2D NMR experiments, still remains to be demonstrated.

CbpD, which is only expressed during competence, consists of 4 domains: the N-terminal
cysteine/histidine-dependent amidohydrolase-peptidase (CHAP) domain that cleaves peptide
bonds of the PGN chain [22,36]; the 2 src-homology 3b (SH3b) domains that recognize and

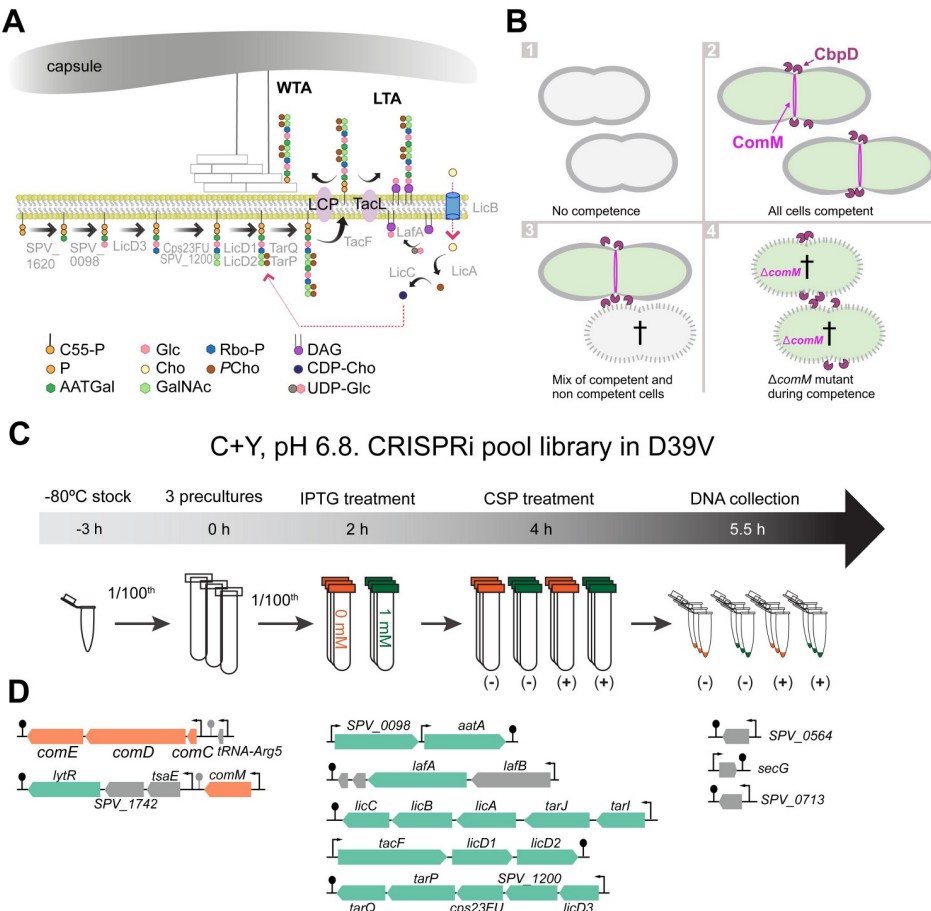

**Fig 1.** (**A**) Proposed teichoic acid biosynthesis pathway in *S. pneumoniae* based on [29–31,55]. Abbreviations: LCP (LytR-Cps2A-Psr family), C55-PP (Undecaprenyl diphosphate), P (Phosphate residue), Glc (Glucose), Rbo-P (ribitol 5-phosphate), DAG (Diacylglycerol), AATGal (2-acetamido-4-amino-2,4,6-trideoxygalactose), Cho (Choline), *P*Cho (phosphorylcholine), CDP-Cho (cytidine diphosphate-choline), GalNac (N-acetylgalactosamine), UDP-Glc (uridine diphosphate glucose). Note that the capsule and the WTA potentially compete for the same anchoring position on the peptidoglycan (O-6 of N-acetylmuramic acid (MurNAc)). (**B**) Pneumococcal fratricide. (1) under non-permissive conditions for competence, D39V cells do not become competent, thus, the fratricin CbpD and the immunity protein ComM are not expressed. (2) Under permissive conditions, all cells become competent [56,57], producing CbpD but also ComM leading to cell elongation [47], thus all cells are protected and there is no fratricide. (3) In conditions where only a subpopulation of cells becomes competent, noncompetent pneumococci lyse by action of CbpD and subsequently release DNA. (4) Deletion of *comM* results in autolysis when competence is activated, via CbpD production. (**C**) Workflow to detect essential genes during competence. Three independent precultures of the CRISPRi pooled library [54] were grown until OD 0.1 in C+Y pH 6.8 to avoid spontaneous competence. Cells were then diluted 100 times in C+Y supplemented with 0 or 1 mM of IPTG. When cells reached $OD_{595\,nm}$ 0.1 again, 100 ng/ml CSP1 was added to the indicated conditions and cells were grown for 2 more generations ($OD_{595\,nm}$ approximately 0.4). Cells were collected, the DNA was isolated, and the library was prepared for Illumina sequencing [53]. (**D**) Operons with fitness cost during competence. In orange, competence-related genes; in green, genes involved in TA biosynthesis, in gray, genes involved in neither competence nor teichoic acid biosynthesis.

bind to PGN; and the C-terminal end consisting of 4 repeating choline-binding sequences that direct CbpD to the septal region of the pneumococcal cell [36]. The initial cell wall damage by CbpD most likely facilitates the accessibility of LytA and LytC, enhancing the fratricide process. Interestingly, several key virulence factors such as CbpG (cleaves host extracellular matrix [37]), pneumococcal surface protein A (PspA—inhibits opsonisation and binding to host lactate dehydrogenase [38]) and PspC (aka CbpA—involved in adherence and complement

evasion [39–42]) are also CBPs [43,44]. Although not understood, a *cbpD* mutant is attenuated in colonization [45].

To avoid committing suicide during fratricide, competent cells produce ComM, an immunity protein [46]. During competence, ComM (ComE-dependent) is produced earlier (approximately 5 min) than CbpD (ComX-dependent). ComM is an integral transmembrane protein, of yet unknown structure, whose role in providing immunity remains to be elucidated. Some clues come from 2 studies that showed that cells become elongated during competence in a ComM-dependent manner, suggesting that ComM regulates septal PGN synthesis [47,48] (Fig 1B).

In addition, competence is also activated when cells are adhering to human epithelial cells [49] and during invasive disease as shown in several mouse and zebrafish models [50–52]. Indeed, competence development is a key pathogenic factor in pneumococcal meningitis [51,52]. However, it is unclear how competence and fratricide play a role in pneumococcal virulence.

Here, we report that TA biosynthesis is essential during competence. We demonstrate that ComM provides immunity against fratricins in concert with LytR, and that LytR is a key enzyme required for WTA anchoring in the pneumococcus. Transcriptional repression of any of the genes involved in WTA assembly will lead to increased susceptibility to CbpD and fratricide. Strikingly, we find that PspA and PspC are localized to the septum and are more readily detected by antibodies when bacteria are competent, with this phenomenon being dependent on CbpD. In line with CbpD-dependent surface exposure of CBPs, pneumococci lacking CbpD show reduced adherence to human nasopharyngeal epithelial cells. The data presented here suggest a model in which competence activation during infection is crucial to expose important virulence factors on the outside of the pneumococcal cell to ensure better adherence to host cells.

## Results

### Teichoic acid biosynthesis is essential during pneumococcal competence as identified by CRISPRi-seq

While it has been shown that the competence-induced ComM protein is required for immunity against fratricins [46], the molecular mechanism underlying immunity remains unclear. To investigate this, we performed a genome-wide CRISPRi-seq screen (Fig 1C and 1D) [53]. We screened a pooled library of pneumococcal strains harboring an inducible dCas9 and a constitutively expressed single-guide RNA (sgRNA), targeting in total 1,498 operons in the genome of strain D39V [54]. The rationale behind this screen being that bacteria undergoing transcriptional down-regulation of genes important for immunity against fratricins will be outcompeted or lysed by fratricins produced by strains in which the sgRNA targets a neutral gene (Fig 1B). The pooled library was grown under competence-permissive and non-permissive conditions by the addition of synthetic CSP1. To confirm that observed fitness costs were due to competence induction and not because of the essentiality of the targeted operon, 4 different conditions were tested (S1A Fig): (I) control (C+Y, pH 6.8; non-permissive for natural competence development); (II) library induction in absence of competence (+ IPTG); (III) competence induction (+ CSP1); and (IV) both library and competence induction (+ IPTG and CSP1). After Illumina sequencing, the fitness of targets and quality of the replicates were then evaluated and the fold change of the abundance of sgRNAs between the 4 groups was analyzed [53] (S1B and S1C Fig).

Fourteen sgRNAs targeting 10 operons were significantly less abundant in condition IV, suggesting that they are essential or become more essential in protecting cells against lysis

during competence (Fig 1D and S2A Fig and S1 Table; a log2FC of <-1, >1, alpha of 0.05). As expected, *comCDE* and *comM* were among the most essential hits, demonstrating that when cells are unable to activate competence or ComM, they are rapidly outcompeted or lysed by competent (and CbpD) producing competitors. Strikingly, most of the other hits were operons related to TA synthesis (Fig 1D).

Next, we examined in more detail the identified sgRNAs targeting genes involved in competence or TA synthesis (S2 Fig). All but 2 operons related to TA synthesis (*SPV_1620-aatB* and *tacL*) were underrepresented (S2A Fig). *SPV_1620-aatB* was less present when competence and the library were induced, although the differences were not significant; however, the sgRNA targeting *tacL*, which transfers TA precursor chains onto the glycolipid membrane anchor (forming LTA) [30,31], did not show any fitness cost during competence (S2B Fig). The 3 remaining sgRNAs targeted *secG* and 2 hypothetical proteins: *SPV_0713* and *SPV_0564*.

## Teichoic acid biosynthesis does not regulate competence development

One hypothesis that could explain why strains carrying a sgRNA targeting a TA gene are underrepresented in the CRISPRi-seq screen is that competence is not activated and hence ComM is not expressed rendering cells susceptible to fratricide. To test whether TAs are required for competence activation, we generated nonpolar depletion strains for each individual gene belonging to the 10 operons that contain the 14 sgRNAs found to become more essential in protecting cells against lysis during competence (S2 Table). The deletions containing an erythromycin resistance cassette were transformed in the corresponding strain harboring an IPTG-inducible copy of the gene ($P_{lac}$ promoter) integrated at the non-essential ZIP locus, in order to control the expression of the genes (e.g., *comM* $^{-/Plac-}$) (S3 Fig). High-content microscopy of the depletion strains showed different phenotypes depending on the targeted gene, as shown before (S4 Fig) [55]. We used a competence-specific induced *ssbB* promoter fused to firefly luciferase ($P_{ssbB}$-*luc*, [56]) to quantify competence activation. Cells were grown in presence of different concentrations of IPTG to not produce or overproduce the TA gene products (0 mM, 0.005 mM, 0.01 mM, 0.1 mM, and 1 mM), and luciferase activity as well as optical density were regularly measured (S5 Fig). As expected, in absence of *comCDE*, competence was not triggered. However, depletion of all tested TA-related genes, except *tacL*, did not affect natural competence development under conditions where cell growth was not compromised (S5 Fig). Interestingly, although the absence of TacL inhibited natural competence, cells remained protected from fratricide when induced by externally added CSP (S6 Fig), explaining why *tacL* depletion did not show any fitness cost in the CRISPRi experiment (S2 Fig).

## Teichoic acid biosynthesis is required to prevent cell lysis by fratricide

As TAs are not required for competence induction (S5 Fig), another hypothesis is that TAs provide resistance to fratricins during competence. In this case, depletion of TA-related genes should result in autolysis by fratricides when competence is triggered. To test this, we used the SYTOX Green Dead Cell Stain (Thermo Fisher Scientific) to evaluate cell lysis in presence and absence of CSP1-induced competence. A *comM* mutant was used as a control strain that should show lysis upon competence induction. Indeed, as shown in S6 Fig, rapid lysis after CSP1 addition was observed for the *comM* mutant, as well as many of the TA-related genes (e.g., *tarP*, *tacF*, *tarI*, *licA*, *licD3*). It was recently shown that the capsule protects pneumococci from LytA-induced lysis [58]. In line with this, depletion of *cps2A* also led to increased competence-dependent lysis, while a *psr* deletion had no phenotype (S6 Fig). Together, this data suggests that, besides ComM, TAs are essential to maintain the integrity of the cells during

"attack" of competence-induced fratricins (i.e., CbpD, LytA). Indeed, it was previously shown that cells depleted for TAs become more susceptible to autolysis by LytA [59].

## *lytR* is in a transcriptional unit with *comM* and is up-regulated during competence by transcriptional read-through

As both CbpD and LytA bind to the choline units on TAs, we focused on the external part of the pathway: the anchoring of TAs into cell wall (WTA) and membrane (LTA), respectively (Fig 2A). TacL is the only known protein in *S. pneumoniae* responsible for anchoring TA chains to a glycolipid, while LCP family proteins (i.e., LytR, Cps2A, and Psr) are the putative proteins anchoring such chains to the cell wall, thereby producing WTA. The fact that *cps2A* and *psr* were not found in the CRISPRi-seq screen (S2 Fig), and that *lytR* forms part of the same operon as *comM* (Fig 2B), suggests a key role for LytR in attaching TAs into the cell wall and subsequent protection from fratricins. Indeed, Ye and colleagues showed that deletion of *lytR* led to decreased retention of TAs to the pneumococcal surface [60]. While capsule mutants are known to be more susceptible to fratricins as the capsule protects the septal cell wall from hydrolysases [58], *cps2A* was not picked up in the CRISPRi-seq screen likely as it takes several generations before cells shed their capsule upon transcriptional down-regulation [61].

While *tacL* is in a single-gene operon, the *comM-tsaE-spv_1742-lytR* operon shows an interesting complexity [62]. Transcription of *comM* is under competence control by ComE, and its imperfect terminator is approximately 61% efficient (Fig 2B). The 3 downstream genes are regulated by an internal transcription start site (TSS) with basal expression. However, when competence is activated, the expression of all 3 genes is highly increased due to inefficient transcriptional termination by the *comM* terminator (Fig 2C; [9,63]). We also note strong activation of *cbpD* transcription and to a lesser extent *lytA* during competence induction, while *tacL* and *lytC* transcription is not changed (Fig 2C). TsaE is a predicted tRNA threonylcarbamoyladenosine biosynthesis protein and SPV_1742 is a predicted acetyltransferase [62]. Individual deletion of *tsaE* or *spv_1742* did not affect cell integrity during competence (Fig 2D and S5 Fig). To further exclude their role in fratricide, we constructed a double Δ*tsaE* Δ*SPV_1742* deletion mutant. As shown in Fig 2D, the Δ*tsaE* Δ*SPV_1742* double mutant was not susceptible to lysis during competence, suggesting they do not play a role in TA synthesis.

## LytR is required for ComM-mediated immunity to CbpD

While *tsaE* and *spv_1742* did not play a role in immunity to fratricins, we were unable to construct a *lytR* deletion strain in the D39V genetic background, suggesting it is an essential gene under our experimental conditions. We note that pneumococcal *lytR* mutants have been successfully constructed before, although serious growth impairments were noted [60,64]. Nevertheless, we were able to construct a complementation strain in which *lytR* is expressed from an IPTG-inducible promoter at the ZIP locus [65] while deleted from its native locus (S3 Fig). To evaluate how LytR protects cells from competence-related lysis, we performed 3 complementary approaches: evaluation of cell lysis in the deletion of the fratricins (Fig 3A), overexpression of *cbpD* (Fig 3B), and addition of exogenous CbpD (Fig 3C). In all 3 approaches, absence of competence or *comM* depletion resulted in a rapid cell lysis when CbpD was present (either induced or added exogenously to the medium). Importantly, depletion of *lytR* strongly increased susceptibility to CbpD (Fig 3C) in line with a previous report [64]. Interestingly, deletion of *cbpD* under *lytR* depletion conditions increased susceptibility to LytA in the stationary phase (Fig 3A). Deletion of both *cbpD* and *lytA* in *lytR* $^{-/Plac-}$ cells does not protect

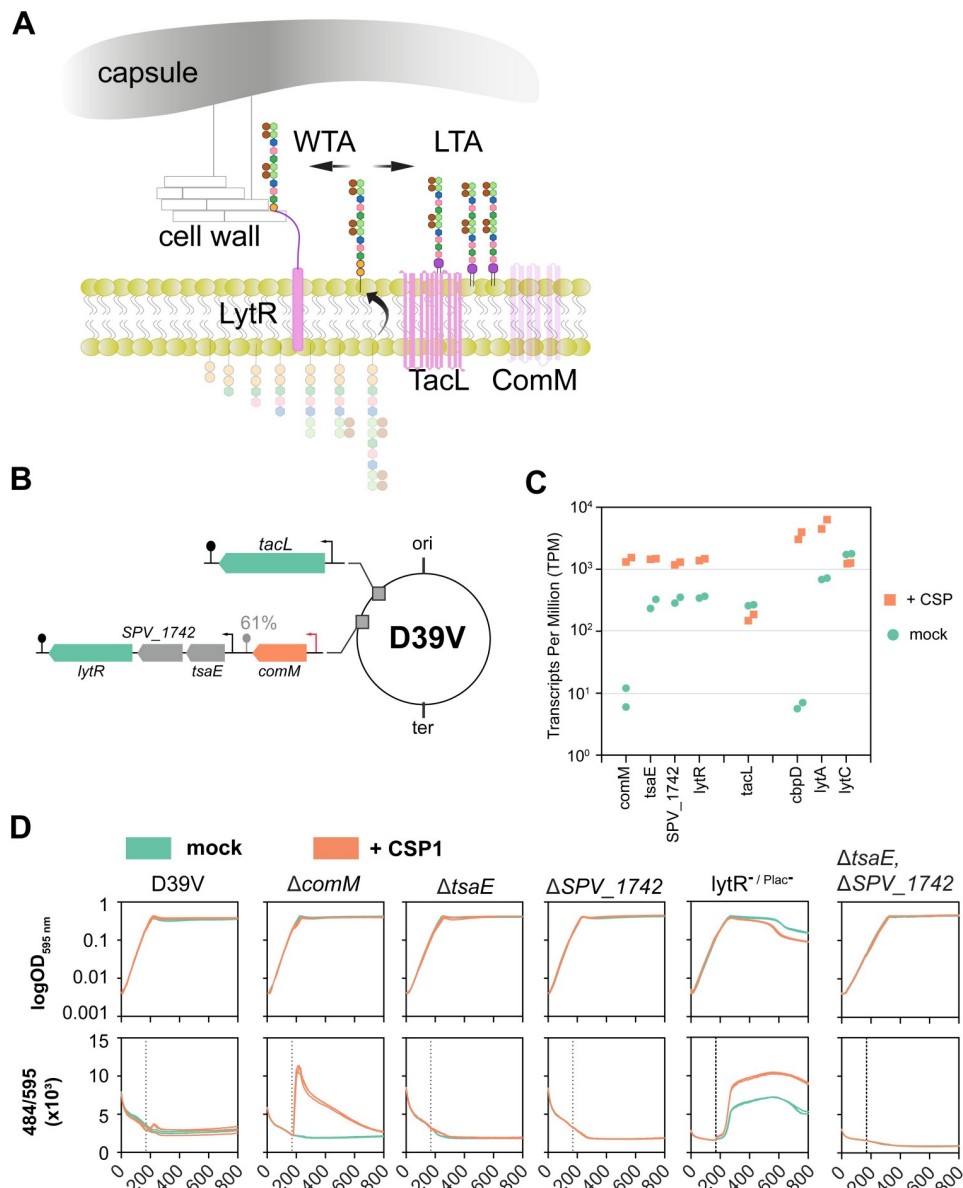

**Fig 2. Role of the *comM-tsaE-SPV_1742-lytR* and *tacL* operons.** (**A**) In *S. pneumoniae* the same TA chains can be anchored to the membrane (by TacL) or to the cell wall with LytR as a key enzyme. (**B**) Schematic representation of the *tacL*, *comM* and *lytR* operons. Note the ComE-activated promoter (orange arrow) upstream of *comM* and the imperfect transcriptional terminator downstream of *comM* [62]. (**C**) Expression levels in competence induced and non-induced cells. RNA-seq data from [9,63]. Duplicates for each condition are shown. (**D**) Cell lysis evaluation. Individual strains were grown in C+Y pH 6.8 to avoid natural competence development in presence of SYTOX Green Dead Cell Stain dye. When cell cultures reached $OD_{595\,nm}$ 0.1 (approximately after 170 min), 100 ng/ml of CSP1 was added to induce competence. Three biological replicates per condition are shown. Since *lytR* is essential in the D39V genetic background [55], we constructed a *lytR* depletion strain in which *lytR* is expressed from the ZIP locus under the IPTG-inducible Plac promoter and deletion from its native locus (lytR $^{-/Plac-}$). This strain was generated in the presence of IPTG and for this experiment grown in the absence of IPTG leading to slow depletion of LytR (raw data in S7 Table). TA, teichoic acid.

them from lysis, confirming the essential role of LytR, not only during competence (Fig 3A). Interestingly, LytR overexpression in the presence of basal levels of ComM showed some level of protection against CbpD-mediated lysis when 0.25 μg/ml of CbpD was added, which was more evident when both ComM and LytR were overexpressed (Fig 3C). Indeed,

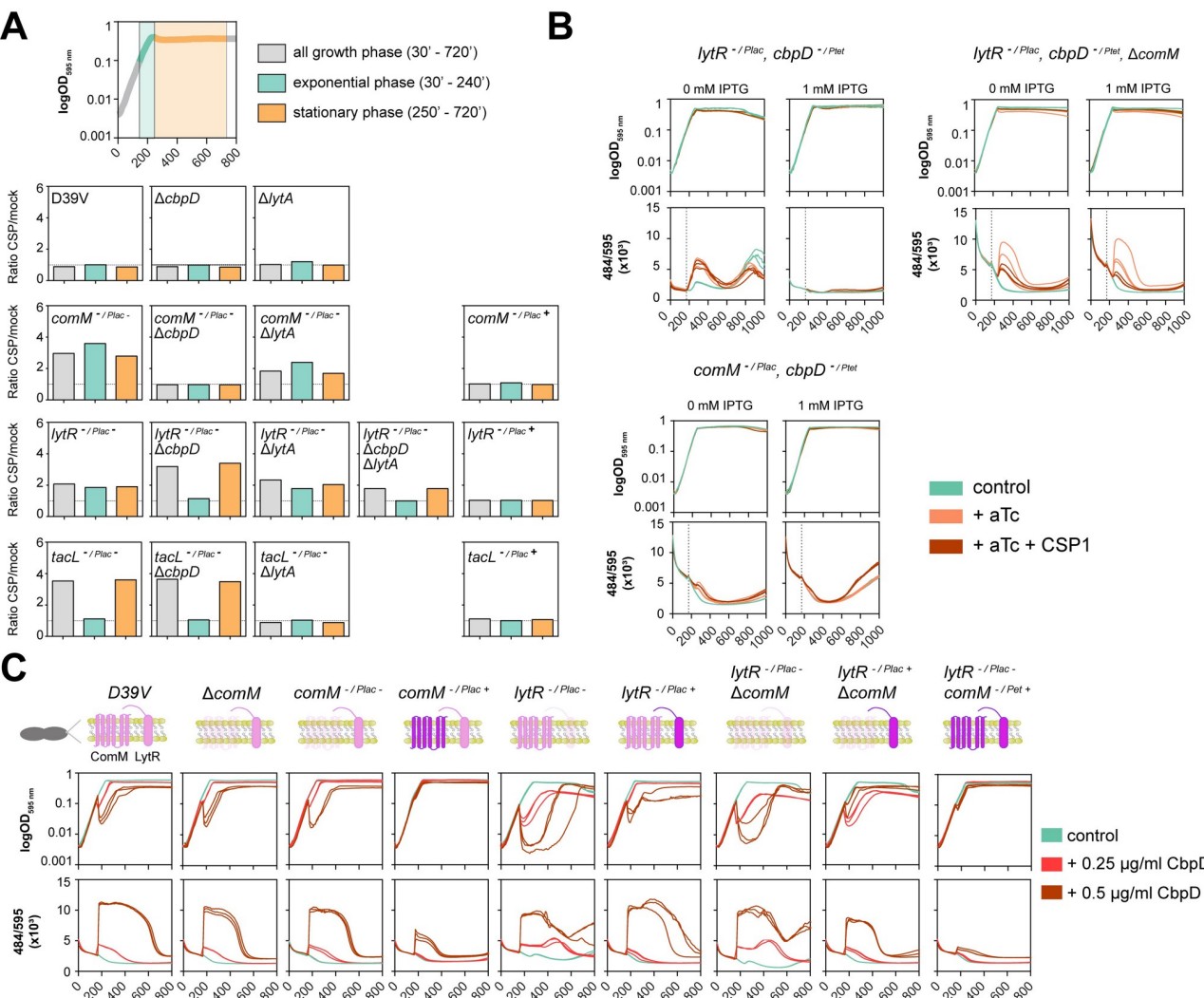

**Fig 3. Both LytR and ComM are required for immunity to CbpD.** (**A**) Cell lysis evaluation in presence and absence of the fratricins. Top: Growth phase was split in 3 periods: exponential phase (green; from CSP1 addition to stationary phase), stationary phase (orange), and all growth data across the curve (gray). Individual strains were grown in C+Y pH 6.8 to avoid natural competence development in presence of SYTOX Green Dead Cell Stain dye. When cell cultures reached $OD_{595 \ nm}$ 0.1 (approximately after 170 min), 0 and 100 ng/ml CSP1 was added to the medium, and the fluorescence ratio (+CSP1/mock) was calculated. Three biological replicates per condition are shown. As fratricide occurs shortly after competence induction, it should be detected in the first period, while autolysis via LytA is detected during stationary phase. Strains containing a complementing copy are indicated by Plac (- indicates no IPTG, + indicates addition of 100 μM IPTG). (**B**) Induced cell lysis by overexpression of *cbpD* (cbpD$^{-/Ptet+}$). Individual strains were grown in C+Y pH 6.8 to avoid natural competence development in the presence of SYTOX Green Dead Cell Stain dye. When cell cultures reached $OD_{595 \ nm}$ 0.1 (approximately after 170 min), 0.5 μg/ml of anhydrotetracycline (aTc; orange) or 0.5 μg/ml of aTc plus 100 ng/ml of CSP1 (red) was added to induce *cbpD* expression and/or competence. Three biological replicates per condition are shown. (**C**) Induced cell lysis by addition of recombinant CbpD (obtained as described in [66]. Individual strains were grown in C+Y pH 6.8 to avoid natural competence development in presence of SYTOX Green Dead Cell Stain dye. When cell cultures reached $OD_{595 \ nm}$ 0.1 (approximately after 170 min), 0.25 μg/ml (red) or 0.5 μg/ml of purified CbpD was added to the medium. Three biological replicates per condition are shown. Top panels show the optical density of the cultures ($OD_{595 \ nm}$) and the bottom panels show the ratio of fluorescence emitted from the dead cells (484 nm) divided by the optical density ($484/595 \times 10^{-3}$) (raw data in S8 Table).

overproduction of both ComM and LytR provided better protection from CbpD-induced lysis than overproduction of ComM alone (Fig 3C). In contrast, native expression and overexpression of LytR in the absence of ComM (*comM* depletion) showed the same levels of lysis as a *comM* mutant, suggesting that both proteins are required to protect from fratricide.

## ComM is important for LytR activity

Competence activation, and subsequent up-regulation of CbpD, ComM and LytR, might change the ratio between LTA and WTA (Figs 1A and 2A) as well as blocking septal PGN synthesis [47] leading to protection from the PGN hydrolytic activity of CbpD. Indeed, cells become elongated when competence is induced and this depends on ComM (S7 Fig, [47,48]). During exponential growth, the action of TacL ensures plentiful production of LTAs [30] (Fig 2A). To investigate this further, we measured the incorporation of TAs in the membrane and cell wall using radioactive Methyl-$^3$H-Choline, as it is taken up and incorporated in TAs [24] (Fig 4A).

Approximately 20 min after competence induction in D39V, cell walls were separated from the membrane fraction (see Methods) and a significant increase in the WTA/LTA ratio was observed (Fig 4B and S3 Table). This shift was reduced when competence was induced in absence of ComM (we used the ΔcomM ΔcbpD double mutant to avoid fratricide, S8A Fig). In this strain, *comM* was replaced by a promoterless kanamycin-resistance cassette in frame, thus, levels of LytR are still highly increased after CSP1 addition. Indeed, an increase in WTA levels

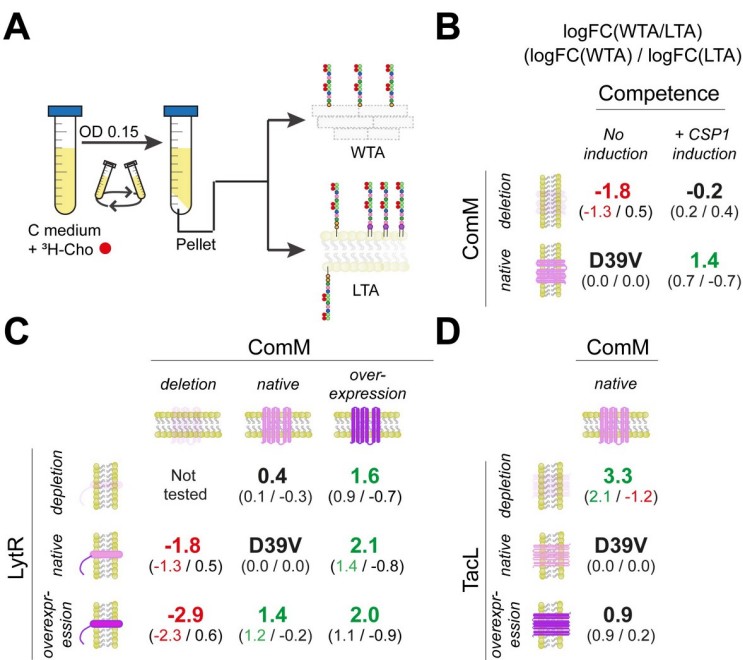

**Fig 4. Competence induction leads to relatively more $^3$H-Cho incorporation in wall TAs compared to lipoteichoic acids. (A)** Protocol to detect $^3$H-Cho. Cells were grown in C medium supplemented with $^3$H-Cho until OD$_{595\,nm}$ 0.15. The pellet was divided in 2 equally sized aliquots, 1 for membrane (including LTAs and TA precursors) and 1 for cell wall isolation (see Methods section for more details). **(B)** $^3$H-Cho detection after competence induction. Membranes and cell walls were isolated to quantify the amount of $^3$H-Cho incorporated. Average of 3 independent biological replicates and 2 technical replicates is shown (raw data in S3 Table). In large the ratio in log fold change between WTA and LTA normalized to D39V wt cells (logFC(WTA0/logFC(LTA) is shown. In small is shown the relative abundancy of WTA or LTA compared to D39V (stated as 0.0/0.0). Red color indicates a significantly smaller ratio between WTA/LTA in that condition compared to wt, whereas green color indicates a significantly greater ratio between WTA/LTA. **(C)** $^3$H-Cho detection in different LytR and ComM conditions. The condition with double *lytR* depletion and *comM* deletion was not tested due to the growth defect and increased lysis of the strain. LytR was depleted for approximately 2 h to not cause too much cell lysis due to its essentiality. Average of 3 independent biological replicates and 2 technical replicates is shown (raw data in S2 Table). **(D)** $^3$H-Cho detection in different TacL conditions. Average of 3 independent biological replicates and 2 technical replicates is shown (raw data in S3 Table). LTA, lipoteichoic acid; TA, teichoic acid; WTA, wall teichoic acid.

was observed in the Δ*comM* Δ*cbpD* mutant strain when competence was triggered; however, those levels are significantly lower compared to D39V (Fig 4B) confirming that ComM is important for WTA production.

To analyze the role of both ComM and LytR in more detail, we tested $^{3}$H-Cho incorporation in strains with different expression levels of both genes (Fig 4C). Interestingly, *comM* overexpression resulted in a reduction of LTA amounts relative to WTA, independent of the *lytR* expression levels (Fig 4C). On the contrary, in absence of *comM* or in its native expression, the amounts of LTA remained similar to those in D39V. When cells were slightly depleted for LytR in native (low) *comM* levels, no significant differences were observed (Fig 4C). In addition, overexpression of LytR also resulted in increased amounts of WTA under basal ComM level conditions, and, for reasons we currently do not understand, a reduction in the WTA/LTA ratio in a *comM* mutant background (Fig 4C and S3 Table).

To complement these results, we also tested the depletion and overexpression of *tacL* (Fig 4D). In absence of TacL, TAs cannot be anchored in the membrane, thereby providing more substrate for LytR to produce WTA (Fig 2A). In support of this model, we find a shifted ratio of WTA/LTA in favor of WTA when TacL was depleted from the cells (Fig 4D). However, in the membrane fraction, we still detected the TAs that are not yet anchored in the cell wall, in addition to the precursors anchored in the inner part of the membrane (Fig 4A). Overexpression of TacL did not impact LTA production during exponential growth, likely because the membrane is already saturated with LTAs in this growth phase [30] (Fig 4D and S3 Table).

## Competence-dependent expression of CbpD exposes PspA and PspC to the cell surface

As shown above, upon competence activation, ComM and LytR alter the pneumococcal cell wall by changing the flux towards WTA biosynthesis and elongating the cell. Indeed, using phase-contrast microscopy, elongation of the cell upon competence induction was diminished in a Δ*comM* mutant (S7A Fig). In addition, we hypothesize that the PGN hydrolase activity of CbpD would result in shedding of the capsule thereby liberating anchor sites on the PGN to which WTA could be attached by LytR. This would imply that CBPs, normally bound to the LTA, now have more binding sites available on the choline residues present on the WTA. To test whether CBPs indeed become more surface exposed during competence development, we performed immunostaining followed by fluorescence microscopy using antibodies raised against PspA and PspC. PspA and PspC are CBPs that play various roles during virulence from blocking complement deposition to binding to lactoferrin [67,68]. As shown in Fig 5A, activation of competence by the addition of synthetic CSP significantly increased antibody staining of both PspA and PspC as measured by immunofluorescence microscopy in strain D39V. Similar results were obtained with strain 19F EF3030, a recent clinical isolate [69]. Crucially, this increased signal was abolished in a *cbpD* mutant but not in a *comM* mutant (Fig 5A). Complementary results were observed when over expressing CbpD or ComM. Here, only the increased expression of CbpD led to higher antibody staining of PspA (S7B Fig). Surprisingly, immunofluorescence microscopy showed a clear septal localization for both PspA and PspC (Fig 5B), while ComM genetically tagged with a yellow fluorescent protein also demonstrated an enriched localization pattern at the septum (S7C Fig). This is in line with previous findings, where LytR was previously also shown to be enriched at the septa and CbpD was shown to bind there as well [32,36]. Moreover, overexpression of LytR showed increased PspA and PspC antibody staining upon competence induction relative to WT, while the depletion of LytR abrogated this effect (S7F Fig). Additionally, localization of PspA fused to GFP showed a more uniform cell wall/membrane localization, whose signal did not change during

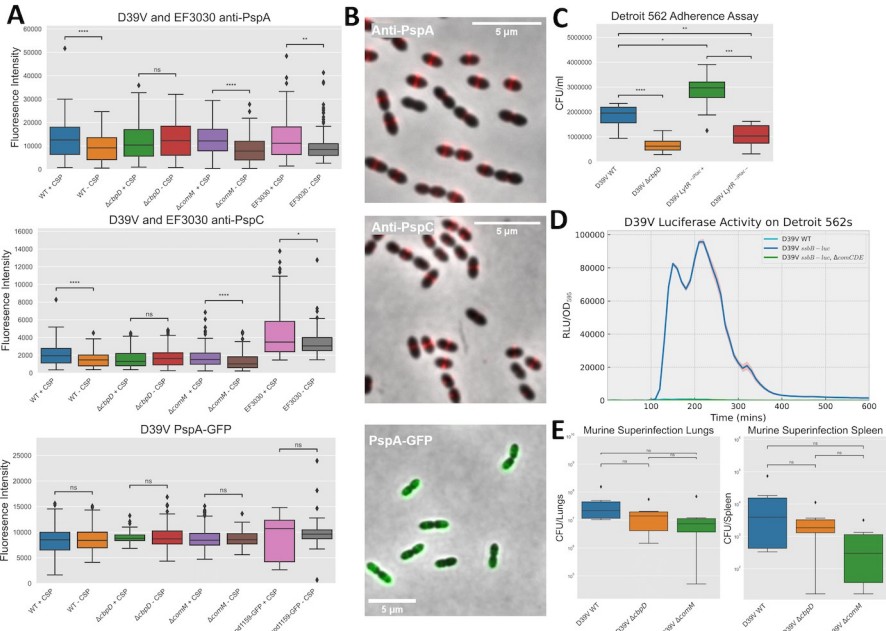

**Fig 5. CbpD-dependent increased surface exposure of PspA and PspC, and impact during infection.** (**A**) D39V (WT, Δ*cbpD* and Δ*comM*) and EF3030 cells, or corresponding D39V strains containing PspA genetically tagged with GFP to its C-terminus, were grown to OD 0.1 in C+Y medium pH 6.8, then exposed to 100 ng/ml or 0 ng/ml of CSP1 for 30 min. For the anti-PspA and anti-PspC microscopy, cells were then stained with primary antibodies raised against PspA or PspC, and then Goat anti-rabbit IgG (H+L) Alexa 555, both at 1/500 dilutions and subjected to epifluorescence microscopy. For the PspA-GFP microscopy, cells were subjected to epifluorescence microscopy directly after exposure to CSP1 for 30 mins. A GFP tagged SPV_1159 D39V strain was used a control as this gene's regulation is not affected by competence induction (Kurushima and colleagues [82]). Fluorescence intensity based on phase-contrast and fluorescence composite images were measured. Diamond symbols represent outlier individual cells. Asterisks show statistically significant differences in fluorescence intensity (ns, not-significant, *, $P < 0.05$, **, $P < 0.01$, and ****, $P < 0.0001$, Mann–Whitney U test) (see Methods section for more details). (**B**) Representative phase contrast and fluorescence composite images taken of D39V WT after anti-PspA and anti-PspC immunostaining, and for D39V PspA-GFP after exposure to CSP1. (**C**) Adherence assay. D39V (WT, Δ*cbpD*, LytR$^{-/Ptet-}$ and LytR$^{-/Ptet+}$) strains were inoculated with Detroit 562 cells to a multiplicity of infection of approximately 20 in RPMI 1640 without phenol red supplemented with 1% (v/v) FBS and 10 mM HEPES buffer for 2 h at 37 ˚C. Cells were then detached, and appropriate dilutions of the cultures were plated on blood agar plates to determine the number of adherent bacteria (see Methods section for more details). Strains containing a complementing copy are indicated by Plac (- indicates no IPTG, + indicates addition of 100 μM IPTG). Data presented are the means ± standard deviation (ns, not-significant, *, $P < 0.05$, **, $P < 0.01$, and ***, $P < 0.001$ ****, $P < 0.0001$, unpaired *t* test). (**D**) D39V (WT, *ssbB-luc*, *ssbB-luc* + Δ*comCDE*) cells were added to Detroit 562 nasopharyngeal cells to an OD$_{595}$ 0.004 in C+Y pH 7.8 +/- 0.05 (permissive conditions for natural competence induction) containing 0.45 mg/ml of luciferine. Cells were incubated in 96-wells microtiter plates with no shaking. Growth (OD$_{595 nm}$) and luciferase activity (RLU) were measured every 10 min for 14 h using a Tecan Infinite F200 PRO. An average of 4 replicates and the ± standard deviation are shown (see Methods section for more details). (**E**) Superinfection with D39V WT, Δ*cbpD* and Δ*comM* mutants. Mice were infected intranasally with 50 PFUs of the influenza A virus strain. Seven days later, mice were inoculated intranasally with 5x10$^{4}$ CFU of *S. pneumoniae* strain D39V, Δ*cbpD* or Δ*comM* mutants. Mice were sacrificed 24 h post-infection and lungs and spleen were sampled to determine the bacterial viable counts. Data presented are the means ± standard deviation (ns, not-significant, unpaired *t* test) from 6 replicates (raw data in S5 and S6 Tables). PFU, plaque-forming unit.

competence (Fig 5A). In corroboration, immunofluorescence microscopy using an unencapsulated mutant displayed less septal localization and more background fluorescence (S7D Fig). These results suggest that while PspA is present throughout the pneumococcal cell surface, it is mainly surface exposed at the newly synthesized septum (Fig 6). Moreover, these data indicate that CBPs become more surface exposed during competence development and that this depends on the activity of CbpD. In line with these results, a *cbpD* mutant has been shown to be significantly attenuated in its ability to colonize the nasopharynx of rats [45]. Moreover,

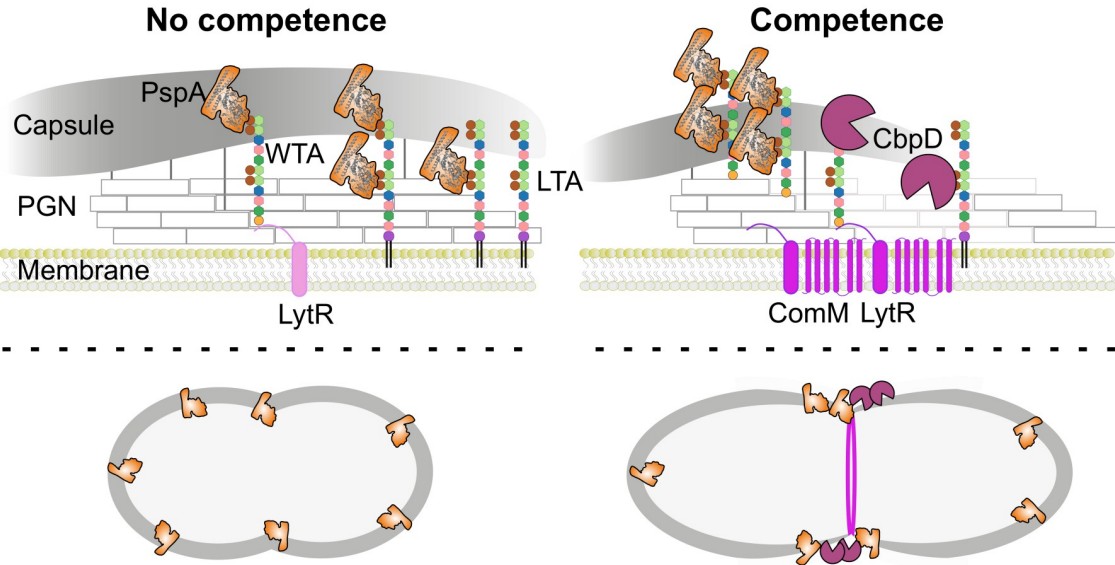

**Fig 6. Model for competence-dependent surface exposure of key virulence factors.** During normal growth, basal levels of LytR support growth by the anchoring of TA precursor to PGN to produce WTAs (left situation). Cells are fully covered by the capsule under noncompetent conditions. Note that the choline-containing WTA is competing with the capsule for binding to PGN at the same sites on the PGN. Consequently, CBPs, including virulence factor PspA, are not fully surface exposed [70]. During competence (right situation), ComM and CbpD are produced, as well as more LytR that is located in the same operon as *comM* (Fig 2). Activation of competence results in cell elongation through ComM localized at the septum [47]. Together, this results in a shift in presence of WTA vs. LTA, providing relatively more choline residues available for binding by PspA within the cell wall (Fig 5). The process is augmented by cleavage of PGN at midcell by CbpD, but cell lysis is prevented through ComM. Together, this results in thinning of capsule at midcell, further exposing PspA to the bacterial cell surface. This model could explain why CbpD mutants show reduced colonization and why competence mutants in general are less virulent (see Discussion). CBP, choline-binding protein; PGN, peptidoglycan; PGN, peptidoglycan; WTA, wall teichoic acid.

adherence assays of D39V WT and corresponding *cbpD* deletion mutant indicated that absence of CbpD attenuated pneumococcal adherence to human nasopharyngeal Detroit 562 cells (Fig 5C), but not for A549 lung epithelial cells (S7E Fig). Additionally, overexpression of LytR increased nasopharyngeal adherence relative to WT, while depletion of LytR led to an abrogation of adherence (Fig 5C), suggesting that CBP surface exposure impacts nasopharyngeal adherence ability of pneumococci. A luciferase assay using a competence-specific reporter demonstrated that competence is indeed activated in the Detroit 562 adherence assay (Fig 5D). Moreover, in a murine influenza A virus superinfection model, we showed that both *cbpD* and *comM* deletion mutants were not significantly attenuated during infection in the lungs or spleen (Fig 5E). This suggests that CbpD is important for colonization but not strictly during replication in the context of flu. These data support a model in which competence is important for pneumococcal adherence by shuttling key virulence proteins, the CBPs, to the outside surface of the cell to interact with the host (Fig 6).

## Discussion

In our study, we identified genes that become essential for protection against lysis during competence in the human pathogen *S. pneumoniae*. Using CRISPRi-seq, we found 14 sgRNAs (targeting 10 operons) underrepresented during competence. Silencing of 2 competence-related operons, *comCDE* and *comM*, showed an expected fitness cost as the immunity protein ComM cannot be produced by down-regulation of both operons, and thus fratricins produced by competing bacteria in the pool lyse these cells. To the contrary, inactivation of the *comAB*

operon did not show any fitness cost as cells can still sense exogenous CSP produced by competing cells and activate competence (S2B Fig).

Interestingly, most of the targeted operons were related to the TA synthesis pathways, suggesting their important role in protecting cells from competence-related autolysis (Fig 1B). Other genes known to play a role in cell wall synthesis were also underrepresented during competence development but did not make the statistical cutoff (S1 Table). This suggests that perturbations in general cell wall homeostasis also sensitizes pneumococci to fratricide. Indeed, it was previously shown that cells depleted for *pbp2b*, *rodA*, *mreD*, *divIVA*, and *cozEa* are more susceptible to lysis by CbpD [71]. Contrary to other gram-positive bacteria, the TAs biosynthesis pathway in *S. pneumoniae* is identical for LTAs and WTAs [29,72]. Using radiolabeled choline, we show that the levels or relative amounts of LTA and WTA are altered when competence is triggered.

At the heart of immunity towards lysis by CbpD lies ComM (Fig 1A and 1C). Here, we show that ComM works in concert with LytR, and expression of both genes provides optimal immunity towards CbpD (Fig 3C). This work also provides strong evidence that LytR is the key enzyme of the LCP-protein family member that anchors the TA precursor to PGN to create WTA (Fig 4). Based on these data, we propose a model in which the flux of TA from LTA increases towards WTA during competence development by the up-regulation of *comM* and *lytR* (Fig 6). This will aid de novo formed CBPs to bind to nascent WTA at the midcell. Alternatively, or at the same time, we hypothesize that the action of CbpD will result in shedding of the capsule thereby liberating WTA binding sites on the PGN. Assuming that CBPs bound to WTA are more surface exposed compared to CBPs bound to LTA, this leads to the model that the competence-dependent remodeling of the cell wall leads to increased surface exposure of CBPs that include major virulence factors such as PspA and PspC (Fig 6). Using immunofluorescence, we show that both PspA and PspC are surface exposed mainly at the midcell, both in serotype 2 strain D39V and in a serotype 19F strain (Fig 5). In this respect, it is interesting to note that CbpD, ComM, and LytR also show septal localization (Fig 5C and [32]), suggesting that the cell wall is predominantly remodeled at the septum during competence and that these provide the highest affinity sites for CBPs. Whether these altered TA fluxes are indirect effects of general ComM-mediated cell wall remodeling, through for instance StkP [47], or that the major function of ComM is to redirect TA to WTA and that this causes upstream effects on septal PGN synthesis remains to be determined. Interestingly, it was recently shown that CbpD only attacks PGN that is newly synthesized by PBP2x and FtsW [66]. Thus, increased WTA-levels (mediated by ComM and LytR during competence) could somehow inhibit the activity of PBP2x or other essential components of the divisome, thereby making the cells resistant to CbpD. This idea is also in line with the observation that noncompetent pneumococci depleted for TacL, and thus with increased levels of WTA (Fig 4D; [30]), are resistant to fratricide. The subsequent delay in division might be important to resolve chromosome dimers that can occur during transformation [47].

Why would competence for genetic transformation have an impact during virulence? Having access to new DNA is not of direct benefit during acute infection. It is tempting to speculate that cell surface remodeling via CbpD-ComM-LytR is one of the main reasons that competence development is switched on during infection. If even a small pool of the CBPs, many of which are bona fide virulence factors such as CbpN (aka PcpA), CbpC, CbpG, PspA, and PspC [43,45,67,68] become more surface exposed, then this might allow bacteria to better adhere or evade the host's immune system (Fig 6). In addition, an increased flux in WTA would also imply reduced capsule anchoring, further improving adherence and surface exposure of the CBPs [68,73]. Whether competence development indeed affects capsule shedding at midcell remains to be tested. It was previously shown, using monoclonal antibodies in

combination with FACS, that certain domains of PspA are surface exposed regardless of the presence of capsule [70]. It could be that the polyclonal PspA antibodies used here have higher affinity for parts of the protein closer to the inner membrane. Regardless, our work shows that PspA and PspC are more surface exposed at mid cell during competence.

The impact of CbpD and CBPs in general on virulence is quite compelling, as it has been shown that the deletion of CbpD inhibits biofilm formation [74], while immunization with purified antibodies against a CbpD peptide fragment significantly increased survival of mice challenged intraperitoneally with lethal levels of a virulent serotype 3 pneumococcal strain [75]. Moreover, the addition of a mutated CSP1 peptide that inhibits the development of competence through outcompeting native CSP1, reduced CbpD and LytA expression levels and significantly decreased murine mortality after pneumococcal intranasal infection [76].

Altogether, the data presented here could explain why CbpD mutants were observed to be attenuated in adherence to human nasopharyngeal cells (Fig 5C) and colonization of the rat nasopharynx [45]. Interestingly, a *cbpD* mutant was not altered in its ability to adhere to A549 lung epithelial cells, likely because adherence is already poor on these cells with wild-type bacteria (S7E Fig). Moreover, while a *cbpD* mutant was not attenuated in its ability to replicate in a mouse viral superinfection model (Fig 5E), *comCDE* mutants were shown to be attenuated in a mouse and zebrafish meningitis model. The key difference between these models is that in the superinfection model, bacteria can immediately replicate without a clear bottleneck imposed by the host immune system [54], while the meningitis models take significant time from inoculation to colonization to dissemination [51,52]. It will be interesting to see how competence and ComM-LytR regulated WTA synthesis plays a role in other disease models as well as in other clinical strains.

## Methods

### Experimental model and subject details

**Bacterial strains and growth conditions.**   All pneumococcal strains used in this study are derivatives of the clinical isolate *S. pneumoniae* D39V [62], Public Health England NCTC14078. Bacterial strains are listed in S2 Table. Growth conditions of bacterial cells were described previously [62]. Briefly, *S. pneumoniae* was grown in C+Y medium (pH 6.8, non-permissive conditions for natural competence induction) at 37˚C and stored at –80˚C in C+Y with 14.5% glycerol at $OD_{595 nm}$ of 0.4.

### Method details

**Competence assays.**   Competence development was monitored in strains containing a transcriptional fusion of the firefly luciferase gene (*luc*) with the late competence gene *ssbB*. The *S. pneumoniae* strains were cultured in a Tecan Infinite F200 PRO allowing for real-time monitoring of competence induction in vitro. A pre-culture was diluted to initial $OD_{595nm}$ 0.004 in C+Y pH 7.8 +/- 0.05 (permissive conditions for natural competence induction) containing 0.45 mg/ml of luciferine and then incubated in 96-wells microtiter plates with no shaking. Growth ($OD_{595 nm}$) and luciferase activity (RLU) were measured every 10 min for 14 h. Expression of the *luc* gene (only if competence is activated) results in the production of luciferase and thereby in the emission of light [77]. An average of 3 replicates and the standard error of the mean (SEM) are shown unless indicated.

**Pooled CRISPRi library.**   The pooled CRISPRi library containing 1,498 sgRNAs targeting all the TSSs in *S. pneumoniae* D39V strain [78] was used to identify essential genes during competence. A pre-culture of the pooled library was grown at in C+Y medium at pH 6.8 to avoid natural competence induction, at 37˚C until $OD_{595 nm}$ of 0.1 (Fig 1A). Then, the

pre-culture was diluted at $OD_{595\,nm}$ 0.005 in presence or absence of 1 mM IPTG to induce the expression of *dCas9*. After 2 h ($OD_{595\,nm}$ approximately 0.1) of library induction, 100 ng/ml CSP1 was added in half of the samples to induce competence. Cells were incubated at 37˚C until $OD_{595\,nm}$ 0.4, and DNA was isolated using a commercial kit (Promega). Preparation of the Illumina library by one-step PCR with commercial oligos and library sequencing were performed following the manufacturer instructions as described previously [54]. The 20 base-pairing sequences of the amplicons were trimmed, mapped, and counted with 2FAST2Q [79]. The count data of sgRNAs were then analyzed with the DESeq2 package in R for evaluation of fitness cost of each sgRNA as described previously [53]. We tested against a log2FC of 1, with an alpha of 0.05.

**Construction of inducible strains.** For every gene related to TA synthesis, an ectopic IPTG-inducible strain was created using the pASR130 (pPEPZ::P*lac*-TmpR-blaR) plasmid that integrates at the non-essential ZIP locus [53]. Primers used to amplify the genes are listed in S4 Table. PCR products were digested with BsmBI, BsaI, or SapI (depending on the presence of restriction sites incompatible with the cloning) and were ligated with similarly digested pPEPZ plasmid containing the P*lac* promoter. The ligation was transformed into strain ADP95 (D39V, *prs1*::P*F6*-*lacI*-*tetR*, *bgaA*::P*ssbB*-*luc*) [65]. All transformants were selected on Columbia blood agar containing 10 μg/ml of trimethoprim, and correct colonies were verified by PCR and sequencing.

**Construction of the native deletions.** Every gene related to TA synthesis was replaced by an erythromycin-resistant marker, in frame with the rest of the genes in the same operon (without promoter and terminator). Primers used to amplify the genes are listed in S4 Table. Upstream region (approximately 1 Kb), downstream region (approximately 1 Kb), and the promoterless erythromycin marker were digested with the indicated restriction enzyme (BsmBI or BsaI, depending on the presence of restriction sites incompatible with the cloning) and were ligated. The ligation was transformed into the corresponding strain carrying the ectopic inducible construct. All transformants were selected on Columbia blood agar containing 0.5 μg/ml of erythromycin and 1 mM IPTG, and correct colonies were verified by PCR and sequencing.

***In vivo* radiolabelling of TA.** *S. pneumoniae* strains (D39V derivatives) were grown in C medium at pH 6.8 (to avoid natural competence development) until $OD_{595\,nm}$ of 0.1 in absence of the inducers. The pre-cultures were diluted to initial $OD_{595\,nm}$ of 0.001 in 10 ml of C medium at pH 6.8 and supplemented with 43.7 mM of $^3$H-choline. Final concentrations of 1 mM of IPTG and/or 500 ng/ml of aTc were added when indicated. Cells were grown at 37˚C without shaking until $OD_{595\,nm}$ of 0.15. For the conditions that required competence induction, 100 ng/ml of CSP1 was added at $OD_{595\,nm}$ 0.10 and incubated for 20 min at 37˚C. Cells were centrifuged for 5 min at 7,000 × *g* and 2 ml of supernatant was collected and stored at –80˚C. The pellet was resuspended in 2 ml of 50 mM MES and distributed in 2 fractions. Membrane and cell wall isolates were obtained as described before [78]. Each growth condition was performed with experimental triplicates.

**Radioactivity quantification.** Radioactivity of cell wall and membrane isolates was measured by scintillation counting using a Liquid Scintillation Analyzer Tri-Carb 4910 TR (PerkinElmer). To do this, 2.5 ml of Ultima Gold XR LSC Cocktail (PerkinElmer, Waltham, Massachusetts) were mixed either with 100 μl of isolated membrane or 200 μl of cell wall isolate. Each experimental triplicate was measured twice. For background correction, 500 μl of cell culture supernatant was treated and measured in the same way.

**Immunostaining followed by microscopy.** *S. pneumoniae* cells were grown in C+Y medium pH 6.8 at 37˚C or in RPMI1640 at 37˚C. For microscopy: Upon reaching $OD_{595} = 0.1$, 100 ng/ml of CSP1 and/or 0.5μg/ml aTc was added and cells were left for 30 min at 37 ˚C

before centrifugation at 10,000 g for 3 min at RT. Cells were then washed with 1 ml of 1× PBS at RT before incubation with PspA or PspC antibodies at a 1/500 dilution in 100 μl 1× PBS at 37 ˚C for 1 h. Cells were again centrifuged at RT and washed with 1 ml of 1× PBS before incubation with 1/500 diluted Goat anti-rabbit IgG (H+L) Alexa 555 (Thermo Fisher Scientific) in 100 μl 1× PBS at 37 ˚C for 1 h. Cells were then washed twice with 1 ml of 1× PBS at RT and resuspended in 50 μl of 1× PBS. Approximately 0.4 μl of cells were subsequently spotted onto PBS-polyacrylamide (10%) pads within a gene-frame (Thermo Fisher Scientific) and sealed with a cover slip as described previously [30]. Microscopy acquisition was performed using a Leica DMi8 microscope with a sCMOS DFC9000 (Leica) camera and a SpectraX light source (Lumencor). Phase-contrast images were acquired using transmission light (100 ms exposure) and still fluorescence images were acquired with 700 ms exposure. The Leica DMi8 filters set used were as followed: YFP (Ex: 500, Dc: 520, Em: 535), Alexa 555 (Ex: 550, Dc: 570, Em: 576), and GFP (Ex: 470/40 nm Chroma ET470/40x, BS: LP 498 Leica 11536022, Em: 520/40 nm Chroma ET520/40 m). Images were processed using LasX v.3.4.2.18368 (Leica). All microscopy images were processed using FIJI v.1.52q (fiji.sc). Calculation of fluorescence intensity based on phase-contrast and fluorescence composite images, as well as cell length measurements, were performed using MicrobeJ [80].

**Detroit 562 adherence and competence assays.** Detroit 562 nasopharyngeal and A549 lung epithelial cells were grown in Dulbecco's Modified Eagle Medium (DMEM) supplemented with 10% fetal calf serum (FCS), 25 mM HEPES buffer (Sigma-Aldrich), and 5 ml of Penicillin-Streptomycin (5,000 U/mL) (Gibco), in 75 cm2 tissue culture flasks (Corning) at 37 ˚C in a 5% CO2 atmosphere. Approximately 1 ml of $4 \times 10^5$ of Detroit 562 or $2 \times 10^5$ A549 cells in the DMEM media were seeded into each well of a 24-well tissue culture tray (Corning Costar) and incubated at 37 ˚C, 5% CO2 for 24 h. Cells were then washed with 1 ml 1× PBS and 1 ml of RPMI 1640 medium without phenol red (Gibco) supplemented with 10% FCS for each well was added. The following day, pneumococcal strains VL1 and VL561 were grown to an OD of 0.1 in C+Y media and 0 ng/ml or 100 ng/ml of CSP1 was added followed by an incubation step of 30 min. Bacteria were then spun down and resuspended in infection medium, RPMI 1640 without phenol red supplemented with 1% (v/v) FBS and 10 mM HEPES buffer. Bacterial samples were added to wells in the 24-well tray containing cells to a multiplicity of infection of approximately 20 and incubated at 37 ˚C, 5% CO2. After 2 h, samples were washed twice with 1 ml 1× PBS, and cells were detached from the plate by treatment with 100 μl of 15 mM sodium citrate and 400 μ 0.1% Triton x-100. Appropriate dilutions of the cultures were plated on blood agar plates to determine the number of adherent bacteria. Assays were performed in quadruplicate from 2 independent experiments. For the pneumococcal D39V competence assay on nasopharyngeal cells, each well of a 96-well tray (Corning Costar) was seeded with 200 μl of $5 \times 10^5$ of Detroit 562 cells in the DMEM media and incubated at 37 ˚C, 5% CO2 for 24 h. Cells were then washed with 200 μl 1× PBS and 200 μl of RPMI 1640 medium without phenol red (Gibco) supplemented with 10% FCS for each well was added. The following day, the competence assay was performed as described above on D39V WT and corresponding strains containing an *ssbB-luc* and/or *cbpD* deletion mutations. An average of 4 replicates and the standard deviation (SD) are shown.

**Murine superinfection.** Male C57BL/6JRj mice (8 weeks old) (Janvier Laboratories, Saint Berthevin, France) were maintained in individually ventilated cages and were handled in a vertical laminar flow cabinet (class II A2, ESCO, Hatboro, Pennsylvania). All experiments complied with national, institutional, and European regulations and ethical guidelines, were approved by our Institutional Animal Care and Use guidelines (D59-350009, Institut Pasteur de Lille; Protocol APAFIS#16966 201805311410769_v3) and were conducted by qualified, accredited personnel. Mice were anesthetized by intraperitoneal injection of 1.25 mg (50 mg/

kg) ketamine plus 0.25 mg (10 mg/kg) xylazine in 200 μl of PBS. Mice were infected intranasally with 30 μl of PBS containing 50 plaque-forming units (PFUs) of the pathogenic murine-adapted H3N2 influenza A virus strain Scotland/20/74 [81]. Seven days later, mice were inoculated intranasally with $5 \times 10^4$ CFU of *S. pneumoniae* strain in 30 μl of PBS. Mice were sacrificed 24 h post-infection by intraperitoneal injection of 5.47 mg of sodium pentobarbital in 100 μl PBS (Euthasol, Virbac, France). Lungs and spleen were sampled to determine the bacterial load. Tissues were homogenized with an UltraTurrax homogenizer (IKA-Werke, Staufen, Germany), and serial dilutions were plated on blood agar plates and incubated at 37˚C. Viable counts were determined 24 h later.

## Quantification and statistical analysis

Data analysis was performed using GraphPad Prism, Microsoft Excel, R version 2.15.1 and RStudio Version 1.0.136.

Data shown in plots are represented as mean of at least 3 replicates ± SEM, as stated in the figure legends. Exact number of replicates for each experiment are enclosed in their respective figure legends.

## Supporting information

**S1 Fig. Evaluation of fitness cost during competence using CRISPRi pool screen.** (A, B) Variance of the replicates. (C) IPTG main effect in competence by interaction of *p*-values. Tested against fold change of 2 (green lines) with alpha of 0.05 (red line) (raw data in S1 Table).
(DOCX)

**S2 Fig. Normalized counts of sgRNAs related to competence and TA synthesis.** (A) sgRNAs with a significant fitness cost during competence. (B) Other sgRNAs related to competence or TA synthesis with no fitness cost. Fitness cost was evaluated as described before (de Bakker and colleagues) [53] (see Methods for more details) (raw data in S1 Table).
(DOCX)

**S3 Fig. Design of inducible systems.**
(DOCX)

**S4 Fig. Morphological changes were examined with fluorescence microscopy, and representative micrographs are shown.** Phase contrast, DAPI staining, and Nile red staining are displayed.
(DOCX)

**S5 Fig. Natural competence development in strains depleted for competence and teichoic acid related genes.** Every strain contains a depletion system by ectopically expressing the indicated gene under control of the Plac IPTG-inducible promoter and the deletion of the gene from its native location. Top, growth curves in absence of the protein (blue = No IPTG) or with different IPTG concentrations. Bottom, area under the curve of relative luminescence values (RLU) of a P*ssbB-luc* construct. For typical P*ssbB-luc* profiles, see, e.g., Slager and colleagues [56]. Note that in absence of *comCDE*, no signal is visible in this kind of data visualization (e.g., see panel 2 Fig 1A, NO IPTG) (raw data in S9 Table).
(DOCX)

**S6 Fig. Detection of cell lysis.** Individual strains were grown in C+Y pH 6.8 to avoid natural competence development in presence of SYTOX Green Dead Cell Stain dye. When OD595 nm reached approximately 0.1, 100 ng/ml of CSP1 was added to induce competence. For those

genes showing less cell lysis in our setup, experiment was performed in absence of IPTG (green and orange). For those essential with severe growth defect, 10 μM IPTG was added to maintain a mild expression of the protein and avoid cell lysis in absence of CSP due to the essentiality of the gene (blue and pink) (raw data in S10 Table).
(DOCX)

**S7 Fig. Cell length measurements and PspA overexpression immunofluorescence after competence induction, PspA immunofluorescence of a Δ*cps* mutant, adherence assay and PspA and PspC immunofluorescence with LytR overexpression or depletion.** (A) D39V cells were grown to OD 0.1 in C+Y medium pH 7.4, then exposed to 100 ng/ml or 0 ng/ml of CSP1 for 30 min. Samples were subjected to epifluorescence microscopy, where phase contrast images were used to measure cell length. Diamond symbols represent outlier individual cell length measurements. Asterisks show statistically significant differences in cell length (Mann–Whitney U test) (see Methods section for more details). (B) D39V strains containing aTc inducible promoters coupled to *cbpD* or *comM* were grown to OD 0.1 in C+Y medium pH 7.4, then exposed to 100 ng/ml or 0 ng/ml of CSP1 for 30 min and/or 100 ng/ml aTc. Cells were then stained with primary antibodies raised against PspA and then Goat anti-rabbit IgG (H+L) Alexa 555, both at 1/500 dilutions and subjected to epifluorescence microscopy. Fluorescence intensity based on phase-contrast and fluorescence composite images was measured. Diamond symbols represent outlier individual fluorescence measurements. Asterisks show statistically significant differences in cell length (Mann–Whitney U test) (see Methods section for more details). (C) Representative phase contrast and fluorescence composite image taken of D39V strain with *comM* genetically tagged with YFP at its C-terminus. (D) D39V Δ*cps* cells were grown to OD 0.1 in C+Y medium pH 7.4, then exposed to 100 ng/ml (left image) or 0 ng/ml (right image) of CSP1 for 30 min. Cells were then stained with primary antibodies raised against PspA, and then Goat anti-rabbit IgG (H+L) Alexa 555, both at 1/500 dilutions and subjected to epifluorescence microscopy (see Methods section for more details). (E) Adherence assay. D39V (WT and Δ*cbpD*) strains were inoculated with lung epithelial A549 cells to a multiplicity of infection of approximately 20 in RPMI 1640 without phenol red supplemented with 1% (v/v) FBS and 10 mM HEPES buffer for 2 h at 37 ˚C. Cells were then detached, and appropriate dilutions of the cultures were plated on blood agar plates to determine the number of adherent bacteria (see Methods section for more details. Data presented are the means ± standard deviation (ns, not-significant, unpaired *t* test). (F) D39V (LytR $^{-/Plac-}$ and LytR $^{-/Plac+}$) were grown to OD 0.1 in C+Y medium pH 6.9, then exposed to 100 ng/ml or 0 ng/ml of CSP1 for 30 min as well 0 μM or 100 μM IPTG. Strains containing a complementing copy are indicated by Plac (- indicates no IPTG, + indicates addition of 100 μM IPTG). Cells were then stained with primary antibodies raised against PspA or PspC, and then Goat anti-rabbit IgG (H+L) Alexa 555, both at 1/500 dilutions and subjected to epifluorescence microscopy. Fluorescence intensity based on phase-contrast and fluorescence composite images were measured. Diamond symbols represent outlier individual cells. Asterisks show statistically significant differences in fluorescence intensity (ns, not-significant and ****, $P < 0.0001$, Mann–Whitney U test) (see Methods section for more details) (raw data in S5 and S6 Tables).
(DOCX)

**S8 Fig. Cell lysis detection in C+Y and C media.** (A) Cell lysis detection in D39V, Δ*comM* and double Δ*comM*Δ*cbpD* strains. Individual strains were grown in C+Y medium (top) or C medium (bottom) at pH 6.8 to avoid natural competence development in presence of SYTOX Green Dead Cell Stain dye. When cell cultures reached $OD_{595\ nm}$ 0.1 (approximately after 170 min), 100 ng/ml of $CSP_1$ was added to induce competence (orange lines). Three biological

replicates per condition are shown. (**B**) Evaluation of cell lysis in the strains used for the radio-active assay (Fig 4). Cells were grown in the indicated medium in presence of SYTOX Green Dead Cell Stain dye. Three biological replicates per condition are shown (raw data in S11 Table).
(DOCX)

**S1 Table. CRISPRi_Seq.** Log2 fold change data between 1 mM IPTG induced vs. uninduced samples of all sgRNA's from the pooled pneumococcal CRISPRi library (see Methods section for more details).
(XLSX)

**S2 Table. Strains.** List of all strains used in this study.
(XLSX)

**S3 Table. 3H-Cho.** Data from $^3$H-Cho detection assays to determine WTA and LTA levels in different pneumococcal strains (see Methods section for more details).
(XLSX)

**S4 Table. Oligos.** List of all oligos used in this study.
(XLSX)

**S5 Table. Microscopy data.** Average mean fluorescence intensity readings and cell length measurements of different pneumococcal strains from the fluorescence microscopy experiments (see Methods section for more details).
(XLSX)

**S6 Table. Infection and *luc* assays.** Raw data from adherence, mice superinfection, and Detroit 562 luciferase assays (see Methods section for more details).
(XLSX)

**S7 Table. Cell lysis evaluation.** Raw data from the cell lysis evaluation in Fig 2 (see Methods section for more details).
(XLSX)

**S8 Table. Induced cell lysis.** Raw data from the induced cell lysis evaluation in Fig 3 (see Methods section for more details).
(XLSX)

**S9 Table. Natural competence.** Optical density and area under the curve data from the natural competence detection assays in S5 data (see Methods section for more details).
(XLSX)

**S10 Table. Cell lysis detection.** Raw data from the cell lysis evaluation in S6 Fig (see Methods section for more details).
(XLSX)

**S11 Table. Media cell lysis.** Raw data from the cell lysis evaluation using C+Y and C media in S8 Fig (see Methods section for more details).
(XLSX)

## Acknowledgments

We thank Olivier Bützberger for technical assistance, construction of strains, and CRISPRi-seq data collection. We also thank Thomas Kohler (University of Greifswald) for supervision and generating antibodies.

## Author Contributions

**Conceptualization:** Vikrant Minhas, Arnau Domenech, Xue Liu, Leiv Sigve Håvarstein, Jan-Willem Veening.

**Data curation:** Xue Liu.

**Formal analysis:** Arnau Domenech.

**Funding acquisition:** Jan-Willem Veening.

**Investigation:** Vikrant Minhas, Arnau Domenech, Dimitra Synefiaridou, Max Brendel, Gonzalo Cebrero, Charlotte Costa, Mara Baldry.

**Methodology:** Vikrant Minhas, Arnau Domenech, Dimitra Synefiaridou, Daniel Straume, Xue Liu, Jean-Claude Sirard, Camilo Perez, Nicolas Gisch, Sven Hammerschmidt.

**Project administration:** Jan-Willem Veening.

**Supervision:** Leiv Sigve Håvarstein, Jan-Willem Veening.

**Writing – original draft:** Arnau Domenech, Jan-Willem Veening.

**Writing – review & editing:** Vikrant Minhas, Dimitra Synefiaridou, Daniel Straume, Jean-Claude Sirard, Camilo Perez, Nicolas Gisch, Sven Hammerschmidt, Leiv Sigve Håvarstein.

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
