## [Editor Report · Decision Letter 0]

17 Aug 2022

Dear Jan-Willem, 

Thank you for submitting your manuscript entitled "Competence remodels the pneumococcal cell wall providing resistance to fratricide and surface exposing key virulence factors" for consideration as a Research Article by PLOS Biology. Although Paula Jauregui, our Microbiology and Immunology Senior Editor, will be the primary handling editor for your submission, I will do so in the next 10 days while she is out of the office, to prevent unnecessary loss of time.

As I mentioned per email, before we can send your manuscript to reviewers, we need you to complete your submission by providing the metadata that is required for full assessment. To this end, please login to Editorial Manager where you will find the paper in the 'Submissions Needing Revisions' folder on your homepage. Please click 'Revise Submission' from the Action Links and complete all additional questions in the submission questionnaire.

Once your full submission is complete, your paper will undergo a series of checks in preparation for peer review. After your manuscript has passed the checks it will be sent out for review. To provide the metadata for your submission, please Login to Editorial Manager (https://www.editorialmanager.com/pbiology) within two working days, i.e. by Aug 19 2022 11:59PM.

Kind regards,

Nonia

Nonia Pariente, PhD

Editor in Chief

PLOS Biology

npariente@plos.org

on behalf of 

Paula

Editor

PLOS Biology

---

## [Decision Letter · Decision Letter 1]

20 Sep 2022

Dear Dr. Veening,

Thank you for your patience while your manuscript "Competence remodels the pneumococcal cell wall providing resistance to fratricide and surface exposing key virulence factors" was peer-reviewed at PLOS Biology. Your manuscript has been evaluated by the PLOS Biology editors, an Academic Editor with relevant expertise, and by several independent reviewers.

As you will see in the reviewer reports, which can be found at the end of this email, although the reviewers find the work potentially interesting, they have also raised a substantial number of important concerns. Based on their specific comments and following discussion with the Academic Editor, it is clear that a substantial amount of work would be required to meet the criteria for publication in PLOS Biology. However, given our and the reviewer interest in your study, we would be open to inviting a comprehensive revision of the study that thoroughly addresses all the reviewers' comments. Given the extent of revision that would be needed, we cannot make a decision about publication until we have seen the revised manuscript and your response to the reviewers' comments. Your revised manuscript would need to be seen by the reviewers again, but please note that we would not engage them unless their main concerns have been addressed. 

Having discussed the reviews with the Academic Editor and in agreement with the reviewers, we think that the competence story is very good. However, the link to virulence is not made with the same rigor. During the revision you should provide better evidence to link virulence to competence in order for the manuscript to be considered at PLOS Biology.

We appreciate that these requests represent a great deal of extra work, and we are willing to relax our standard revision time to allow you 6 months to revise your study. Please email us (plosbiology@plos.org) if you have any questions or concerns, or envision needing a (short) extension.

**IMPORTANT - SUBMITTING YOUR REVISION**

*Resubmission Checklist*

*Published Peer Review*

*PLOS Data Policy*

*Blot and Gel Data Policy*

Sincerely,

Paula

---

Senior Editor

PLOS Biology

REVIEWS:

Reviewer #1: Elaine Tuomanen. Molecular pathogenesis of Streptococcus pneumoniae invasion and inflammation.

Reviewer #2: Donald Morrison. Molecular biology of genetic recombination, specifically on transformation mechanisms in Streptococcus quorum sensing.

Reviewer #3: Molecular basis of host–pathogen interactions in the human respiratory tract.

Reviewer #1: This is an outstanding manuscript both for clarity as well as for detailed experimental design. The diagrams of the CRISPRi strategy and other figures are extremely helpful and informative. The entire text deftly wades through the complexity of the events around competence and draws on extensive literature to summarize and unify concepts from a confusing array of studies. 

Minor points for clarity:

1) Line 82: The original reference for choline binding proteins attaching to Pcho on teichoic acids is not Galan 2015. Suggestions are

Bean B, Tomasz A. Choline metabolism in pneumococci. J Bacteriol. 1977 Apr;130(1):571-4. doi: 10.1128/jb.130.1.571-574.1977. PMID: 15988

Höltje JV, Tomasz A. Specific recognition of choline residues in the cell wall teichoic acid by the N-acetylmuramyl-L-alanine amidase of Pneumococcus. J Biol Chem. 1975 Aug 10;250(15):6072-6. PMID: 238995.

2) The difference in apparent distribution of Cbps when measured by antibody binding vs GFP is noteworthy. At first it could be potentially explained by increased access of antibody to the growth zone perhaps due to loss of capsule. However, this would not agree with older work from the Briles lab where capsule does not block PspA antibody but does block antiphosphorylcholine antibodies. Please discuss. 

Daniels CC, Briles TC, Mirza S, Håkansson AP, Briles DE. Capsule does not block antibody binding to PspA, a surface virulence protein of Streptococcus pneumoniae. Microb Pathog. 2006 May;40(5):228-33. doi: 10.1016/j.micpath.2006.01.007. Epub 2006 Mar 15. PMID: 16540281. 

3) On occasion it is not clear what statements are based on literature vs representing a hypothesis that would explain current results that are not directly shown.

 a) Line 370/483: As the experiments are presented statements are made that CbpD results in capsule shedding. However, this is not shown experimentally and thus should be hypothesized. Similarly, CbpD hydrolyzes cell wall (details unclear), and it is not known if this results in release of cell wall components. 

 b) Please clarify the literature evidence that Cbps attach preferentially to WTA vs LTA? Or modify language to suggest this as a hypothesis supported by your work.

 c) Please clarify if there is literature evidence that LytR attaches TA to cell wall in addition to the results that are compatible with but do not directly show it in this paper.

Reviewer #2: MS, "Competence remodels the pneumococcal cell wall providing resistance to fratricide and surface exposing key virulence factors" by Havarstein, Veening, and co-authors, reports an important advance in understanding of the role of genetic competence in the pathobiology of this pervasive human pathogen. Du to the pioneering work of one of the senior authors , LSH, we have long understood that a typical temporarily competent pneumococcal culture is a mixture of competent cells and others that have not responded to the endogenous competence pheromone, CSP, remaining non-competent bystanders. Furthermore, the competent cells elaborate, and become immune to, lysins that provoke lysis of some of the non-competent relatives ("fratricide"). Both the full story of the fratricidal killing and the mechanism of the self-immunity established in competent cells themselves have remained elusive. This MS reports design and results of an elegant genome-wide screen for genes that support the immunity phenotype.

Unexpectedly, but with satisfying consistency, hits that abrogated immunity were mapped, nearly uniquely, to genes of the teichoic acid synthesis and assembly pathway. The bulk of the MS follows hypotheses as to how and why the unexpected significance of TA for competence immunity arises.

The MS Is an impressive 'tour de force', with well described experiments throughout. However, as submitted the MS stands in need of careful copy editing to reach the quality expected for a Research Article at PLOS Biology, as suggested by this sampling of some specific apparent issues marked below.

l. 2. Reverse "surface exposing".

l. 107. Why is this most likely?

l. 128. 'here we report'

l. 137. Shouldn't "to" be "on"?

l. 163. This may be just a matter of taste but: the use of "essential" in this context may be misleading. We usually consider that if something essential is missing, the cell will be dead. Unless I misread the results, what is reported/detected here is that cells missing some identified genes are more susceptible to lytic killing, not absolutely "dead in the water."

l. 164. Please define "top".

ll. 169.-175. This text may be too telegraphic. A naïve reader may be confused as to which operons are which.

l. 178. Is this pathway proposed here, or is a citation to its previous proposal in order?

l.202. 10 TA operons?

l. 212. Is this repression or simply gene deletion?

l. 211 (fig 5) Please explain a bit more how these patterns should be understood. For example, what is the significance of ssbB expression very late into stationary phase? Also, where could a reader look to find the actual kinetics of its expression?

l. 225. Please explain the L/D scale. Would % lysed be a clearer metric?

l. 392. How can a del-cps strain be encapsulated? Is there a typo here?

l. 445. Is "subsequently" intended to mean "consequently"?

l. 456. As noted above, "essential" may be an oversimplification.

l. 458. The operons do not "show an expected cost". Rather, it is interfering with these operons that creates a fitness cost.

l. 461. "to the contrary, inactivation of the comAB ….

ll. 463-5. The text seems confused here.

l. 503. Perhaps "required" is too strong.

l. 814. 'as describeed"

LL. 848-853. Is the volume unit "ml" really intended?

l. 855. Note that "to an OD of 0.1" is redundant.

l. 857 and ff. Some steps have a temperature indicated, whilst others are not specified

Reviewer #3: General Comments:

This submission explores the mechanism of immunity against competence-dependent fratricide in Streptococcus pneumoniae. The main results indicate a role of LytR (a key enzyme known to be involved in wall teichoic acid formation) and cell lysis mediated by fratricide proteins that bind phosphocholine on pneumococcal teichoic acid. The association with teichoic acid biosynthesis provides new and interesting insight into the workings of fratricide. A number of questions, outlined below, about this part of the study remain and should be addressed. 

A second part of the study is the relationship between fratricide and virulence. The data within this part of the story regarding the surface exposure of virulence factors is far too preliminary to support the model that competence promotes adherence to host cells (in vitro) and/or virulence (in vivo) by altered exposure of choline binding proteins on the bacterial cell surface. This reviewer suggests that this aspect of the submission be removed entirely until there is sufficient evidence to support the mechanism the authors are proposing.

Specific Comments:

1. Lines 227-228: The authors have shown that ComM and TAs are important to maintain the integrity of cells during competence activation. To show that this process is in fact fratricide mediated, could the authors test whether these genes contribute to lysis in a fratricin-mutant (eg CbpD) background? 

2. Lines 238-241: The authors suggests that genes encoding for other LCP family proteins - cps2A and psr are not important for TA attachment to the cell wall and protection from fratricins. However, authors have not included any data (eg. cps2A and psr mutants) to demonstrate that these genes do not play a role in providing protection from fratricins. In fact, data from Fig S6 shows increased lysis upon competence stimulation for the Cps2A mutant. This would suggest these players are in fact important to maintain the integrity of the cell during fratricide. It would be beneficial to establish whether these proteins play any role in the phenotypes reported in this manuscript. 

3. Lines 285-287 and Fig 3C: In contradiction to the statement, the graph (6th panel in Fig 3C) does not show an increased protection (or less lysis) upon LytR overexpression? 

4. Line 312: It may be more accurate to state that ComM is "important" for LytR activity since increased WTA is produced upon LytR induction in a ComM mutant (Table S3). 

5. Lines 324-325: Fig 4B still shows the shift from a fold change of -1.8 to -0.2.

6. Fig 4C: Why is there a reduction in WTA/LTA ratio upon overexpression of LytR in a ComM mutant? 

7. Line 214: Can you normalize AUC RLU against the growth? To claim that the low RLU under low IPTG treatment is due to growth defect but not inadequate competence induction in order to rule out the possibility that the gene of interest is indeed regulating the competence development, show that after normalization against growth, there is no significant difference among different IPTG treatment groups.

8. Fig. 2D: Shouldn't there also be a condition of lytR-/Plac- with IPTG induction as a complement group of LytR deletion?

9. Line 269: In Fig. 3C, the comM overexpression alone (comM-/Plac+) has relieved the lysis compared to ∆comM or comM-/Plac-. It's likely that ComM alone can deal with CbpD-dependent lysis. You didn't establish that ComM-mediated protection is LytR-dependent by deleting LytR. Suggest testing the condition of comM overexpression in the absence of LytR (comM-/Plac+, lytR-/Plac-). Also, if you compare D39V with lytR-/Plac-, the lysis was even worse in the presence of LytR in D39V.

10. Fig. 3C: Another issue with Fig. 3C is that it's not convincing to interpret the lysis results with the growth defect. Similar to Point 7 mentioned above, is there a way to normalize the lysis against growth?

11. Line 370-372: What is the evidence to support this claim that the capsule is thinned by CbpD?

12. Line 409: Fig. 5: CbpD-dependent increased surface exposure of PspA and PspC is shown, but not any LytR-dependent effect, which made this part of the story off topic from the detailed analysis of TA. More experiments are needed to show the role of LytR and link to fratricins during this process.

13. Line 409: Fig. 5: Data is insufficient to establish causality that the difference in adherence assay (Fig. 5C) is due to different surface exposure of virulence factors. Effects shown in C, E are not shown to be TA dependent. Effects shown in C, E are not shown to explain the 'virulence' effects shown in A. Is adherence and lung superinfection dependent on the display of these CBPs? In general, "competence is crucial for host adherence by increased surface exposure of its various CBPs" is overstated based on the evidence you've shown here.

14. It exactly clear why the LytA-mediated shift from LTA to WTA effects CBPs since both forms of TA are contain phosphocholine? Is the hypothesis that WTA is more exposed than LTA? If so, evidence to support this concept would strengthen the proposed model.

Minor Comments:

1. Line 49-51. It is confusing to state that Spn is a commensal and then describe it as a public health problem in the next line.

2. Line 154 and rest of the manuscript: 1 in CSP1 should not be in subscript. 

3. Line 327: CSP1 and not CPS1

4. Line 166: not "competent producing competitors", but "competence producing competitors"

5. Line 167: Give details in the Methods about your cutoff of the top hits. 

6. Line 169: should be "either competence or TA synthesis" but not "both"

7. Line 185: Even under permissive conditions, a large proportion but not all the cells can become competent?

8. Line 188: This sentence is ambiguous. "Deletion of comM results in autolysis via CbpD production when competence is activated".

9. Line 194-195: didn't mention what's in grey. 

10. Line 220: reference Fig. S5 instead of Fig. S6

11. Line 266-267: This sentence is confusing. Do you want to say as LytR is essential for normal growth, so you used lytR-/Plac- instead of a ∆lytR clean deletion to deplete for LytR? Also, in Fig. S6, you added 10µM IPTG to treat Plac-lytR-/+; but you didn't add IPTG to lytR-/Plac- in Fig. 2D. Is that because lytR-/Plac- doesn't require IPTG to remain viable as Plac-lytR-/+ does? The experimental setup is confusing until the reader reaches lines 270-6.

12. Line 287: reference Fig. 3C instead of Fig. 3

13. Line 381: clarify that the 19F strain is EF3030 annotated in Fig. 5

14. Line 292: In Fig. 3A, should address why the lytR-/Plac- ∆cbpD has more lysis compared to lytR-/Plac-

15. Line 337: reference Fig. 4C instead of Fig. 5C

16. Line 467: "suggesting their important role in protecting cells"

17. Line 807: "grown in C+Y medium at pH 6.8"

18. Line 819: add comma

19. Line 872: "Calculation of fluorescence intensity based on…"

20. Line 884: "before adding 0ng/mL or 100ng/mL CSP"

21. Line 911: "5x104"

---

## [Editor Report · Decision Letter 2]

6 Dec 2022

Dear Dr Veening,

Thank you for your patience while we considered your revised manuscript "Competence remodels the pneumococcal cell wall providing resistance to fratricide and exposes key surface virulence factors" for publication as a Research Article at PLOS Biology. This revised version of your manuscript has been evaluated by the PLOS Biology editors and the Academic Editor.

Based on our Academic Editor's assessment of your revision, we are likely to accept this manuscript for publication, provided you satisfactorily address the following data and other policy-related requests.

1. DATA POLICY:

A) Supplementary files (e.g., excel). Please ensure that all data files are uploaded as 'Supporting Information' and are invariably referred to (in the manuscript, figure legends, and the Description field when uploading your files) using the following format verbatim: S1 Data, S2 Data, etc. Multiple panels of a single or even several figures can be included as multiple sheets in one excel file that is saved using exactly the following convention: S1_Data.xlsx (using an underscore).

B) Deposition in a publicly available repository. Please also provide the accession code or a reviewer link so that we may view your data before publication.

Regardless of the method selected, please ensure that you provide the individual numerical values that underlie the summary data displayed in the following figure panels as they are essential for readers to assess your analysis and to reproduce it: Figures 2CD, 3ABC, 4BCD, 5ACDE, S1ABC, and Supplementary Figures S2AB, S5ABCDEFGHIJK, S6ABCDEFGHIJK, S7ABEF, S8AB.

**Please also ensure that figure legends in your manuscript include information on where the underlying data can be found, and ensure your supplemental data file/s has a legend.**

2. We suggest a modification in the title: "Competence remodels the pneumococcal cell wall exposing key surface virulence factors that mediate increased host adherence".

We expect to receive your revised manuscript within two weeks.

*Published Peer Review History*

*Press*

Sincerely,

Paula

---

Senior Editor,

pjaureguionieva@plos.org,

PLOS Biology

---

## [Editor Report · Decision Letter 3]

4 Jan 2023

Dear Dr Veening,

Thank you for the submission of your revised Research Article "Competence remodels the pneumococcal cell wall exposing key surface virulence factors that mediate increased host adherence" for publication in PLOS Biology. On behalf of my colleagues and the Academic Editor, Alice Prince, I am pleased to say that we can in principle accept your manuscript for publication, provided you address any remaining formatting and reporting issues. These will be detailed in an email you should receive within 2-3 business days from our colleagues in the journal operations team; no action is required from you until then. Please note that we will not be able to formally accept your manuscript and schedule it for publication until you have completed any requested changes.

PRESS

Sincerely, 

Paula Jauregui

---

Senior Editor

PLOS Biology
